# The DNA sensors AIM2 and IFI16 are SLE autoantigens that bind neutrophil extracellular traps

Brendan Antiochos[1]*, Daniela Trejo-Zambrano[1], Paride Fenaroli[2,3], Avi Rosenberg[3], Alan Baer[1], Archit Garg[4], Jungsan Sohn[1,4], Jessica Li[1], Michelle Petri[1], Daniel W Goldman[1], Christopher Mecoli[1], Livia Casciola-Rosen[1], Antony Rosen[1,3]

[1]Johns Hopkins University School of Medicine, Division of Rheumatology, Baltimore, United States; [2]Nephrology Unit, Parma University Hospital, Department of Medicine and Surgery, Parma, Italy; [3]Johns Hopkins University School of Medicine, Division of Pathology, Baltimore, United States; [4]Johns Hopkins University School of Medicine, Department of Biophysics and Biophysical Chemistry, Baltimore, United States

*For correspondence:
bantioc1@jh.edu

Competing interest: The authors declare that no competing interests exist.

## Abstract

**Background:** Nucleic acid binding proteins are frequently targeted as autoantigens in systemic lupus erythematosus (SLE) and other interferon (IFN)-linked rheumatic diseases. The AIM-like receptors (ALRs) are IFN-inducible innate sensors that form supramolecular assemblies along double-stranded (ds)DNA of various origins. Here, we investigate the ALR absent in melanoma 2 (AIM2) as a novel autoantigen in SLE, with similar properties to the established ALR autoantigen interferon-inducible protein 16 (IFI16). We examined neutrophil extracellular traps (NETs) as DNA scaffolds on which these antigens might interact in a pro-immune context.

**Methods:** AIM2 autoantibodies were measured by immunoprecipitation in SLE and control subjects. Neutrophil extracellular traps were induced in control neutrophils and combined with purified ALR proteins in immunofluorescence and DNase protection assays. SLE renal tissues were examined for ALR-containing NETs by confocal microscopy.

**Results:** AIM2 autoantibodies were detected in 41/131 (31.3%) SLE patients and 2/49 (4.1%) controls. Our SLE cohort revealed a frequent co-occurrence of anti-AIM2, anti-IFI16, and anti-DNA antibodies, and higher clinical measures of disease activity in patients positive for antibodies against these ALRs. We found that both ALRs bind NETs in vitro and in SLE renal tissues. We demonstrate that ALR binding causes NETs to resist degradation by DNase I, suggesting a mechanism whereby extracellular ALR-NET interactions may promote sustained IFN signaling.

**Conclusions:** Our work suggests that extracellular ALRs bind NETs, leading to DNase resistant nucleoprotein fibers that are targeted as autoantigens in SLE.

**Funding:** These studies were funded by NIH R01 DE12354 (AR), P30 AR070254, R01 GM 129342 (JS), K23AR075898 (CM), K08AR077100 (BA), the Jerome L. Greene Foundation and the Rheumatology Research Foundation. Dr. Antiochos and Dr. Mecoli are Jerome L. Greene Scholars. The Hopkins Lupus Cohort is supported by NIH grant R01 AR069572. Confocal imaging performed at the Johns Hopkins Microscopy Facility was supported by NIH Grant S10 OD016374.

## Editor's evaluation

This paper identifies proteins that serve as targets of self-reactive autoantibodies during an autoimmune disease called systemic lupus erythematosus (commonly referred to as lupus). Importantly, the authors provide evidence that these proteins bind and protect extracellular DNA from destruction,

and propose that this property may enhance the autoimmune response to the DNA and associated proteins. The work may therefore provide an important underlying mechanism for a prevalent and important human autoimmune disease.

## Introduction

Systemic lupus erythematosus (SLE) is a rheumatic disease characterized by upregulated interferon (IFN) expression and autoantibody production (*Gupta and Kaplan, 2021*). Autoantibodies inform the identification of specific disease phenotypes and also provide insight into the mechanisms operative in rheumatic diseases (*Rosen and Casciola-Rosen, 2016*). Many SLE autoantigens are nucleic acid binding proteins, and nucleic acid containing immune complexes are implicated in aspects of pathogenesis (*Mahajan et al., 2016*).

The AIM2-like receptors (ALRs) are a group of IFN-induced innate sensors of double-stranded (ds) DNA. AIM2 and IFI16 are the most studied members of the ALR family, which also includes IFIX and MNDA. The ALRs bind to dsDNA in a sequence-independent manner via electrostatic interactions with the dsDNA backbone and form an oligomerized filament along areas of accessible dsDNA of any origin (*Morrone et al., 2014*; *Morrone et al., 2015*). These innate sensors equip the cell with a means of identifying harmful stimuli, including viral genomes, mislocalized mitochondrial DNA, and chromosomal DNA from tumor cells. Once activated, the ALRs activate downstream innate immune signaling by type I IFN and inflammasome (IL-1/IL-18) pathways (*Hornung et al., 2009*; *Unterholzner et al., 2010*).

Anti-IFI16 antibodies occur in both SLE and Sjogren's Syndrome (SS), but we have previously reported that the targeted epitopes differ in these diseases (*Antiochos et al., 2018*; *Baer et al., 2016*). IFI16 oligomers appear to be recognized by SS sera, suggesting that dsDNA binding may enhance its antigenicity. While AIM2 assembles similar filamentous structures on dsDNA, its status as an autoantigen has not been reported. Here, we identify AIM2 as an autoantigen in SLE (targeted in 31.3% of patients), with antibodies against AIM2, IFI16, and dsDNA being highly associated with one another. To understand why anti-ALR and anti-dsDNA antibodies might be closely co-targeted in SLE, we considered the possibility that ALRs bind to neutrophil extracellular traps (NETs) in the extracellular space. NETs are microbicidal structures consisting of protein-laden chromatin fibers generated by neutrophils in response to various stimuli (*Brinkmann et al., 2004*). The NET dsDNA scaffold is a structure on which a variety of molecules interact (*Gugliesi et al., 2013*), representing a platform for antigenic materials (including SLE autoantigens) to be presented to the adaptive immune system (*Mistry and Kaplan, 2017*). We find that both AIM2 and IFI16 bind NETs in vitro and in tissues, with their binding yielding polymeric structures that confer resistance to DNase I. Together, our findings demonstrate that AIM2 and IFI16 are NET-bound autoantigens in SLE.

## Materials and methods

### Patients

Plasma from 131 SLE patients (defined by the SLICC criteria *Petri et al., 2012*) in the Hopkins Lupus Cohort was studied for autoantibodies. Sera from 49 healthy controls were analyzed to establish a threshold for assay positivity. 133 primary was analyzed to establish a threshold for assay positivity. 133 primary Sjögren's Syndrome (SS) patients (defined by ACR/EULAR criteria *Shiboski et al., 2017*) were included as disease controls. All patients and healthy controls gave informed consent for blood used in research, and all work involving human subjects was approved by the Johns Hopkins Institutional Review Board. Paraffin sections from SLE renal biopsies were obtained for immunostaining and are detailed in the *Supplementary file 1c*.

### ALR autoantibody assays

Full length AIM2 cDNA was subcloned into the pET28 vector (Novagen) and used to generate $^{35}$S-methionine labeled AIM2 protein by in vitro transcription and translation (IVTT) (Promega). Immunoprecipitations (IP) were performed using IVTT product diluted in Lysis Buffer (20 mM Tris pH 7.4, 150 mM NaCl, 1 mM EDTA pH 7.4, 1% NP40) and 1 microliter of serum (90 min, 4 °C). 20 microliters of Protein G Dynabeads (Thermo Fisher) were then added to each IP and incubated for 60 min. Beads

**eLife digest** Systemic lupus erythematosus (SLE or lupus for short) is an autoimmune disease in which the immune system attacks healthy tissue in organs across the body. The cause is unknown, but people with the illness make antibodies that stick to proteins that are normally found inside the cell nucleus, where DNA is stored. To make these antibodies, the immune system must first 'see' these proteins and mistakenly recognise them as a threat. But how does the immune system recognise proteins that are normally hidden inside cells?

During infection, a type of immune cell called a neutrophil releases DNA from its nucleus to form structures called neutrophil extracellular traps, or NETs for short. The role of these NETs is to capture and kill pathogens, but they also expose the neutrophil's DNA and the proteins attached to it to other immune cells. It is therefore possible that other immune cells interacting with NETs during infection may contribute to the development of lupus. Two proteins of interest are AIM2 and IFI16. These proteins form large, shield-like structures around strands of DNA, and previous work has shown that some people with lupus make antibodies against IFI16.

Antiochos et al. wondered whether IFI16 and AIM2 might stick to NETs, exposing themselves to the immune system. Examining the blood of people with lupus revealed that one in three of them made antibodies that could stick to AIM2. Those people were also more likely to have antibodies that could stick to IFI16 and to strands of DNA. Using microscopy, Antiochos et al. also found AIM2 and IFI16 on NETs in the kidneys of some people with lupus. Further investigation showed that the presence of AIM2 and IFI16 prevents NETs from breaking down.

If proteins like AIM2 and IFI16 can stop NETs from breaking down, they could allow the immune system more time to develop antibodies against them. Further investigation could reveal whether this is one of the causes of lupus. A clearer understanding of the antibodies could also boost research into diagnosis and treatment.

were magnetically isolated, washed, and boiled in gel application buffer. IP products were electrophorezed on SDS-polyacrylamide gels and visualized by fluorography. Films were scanned and AIM2 bands quantified using Quantity One software (Bio- Rad). IP products were normalized to the same positive reference serum included on each gel. The cutoff for antibody positivity was set at 2 standard deviations above the mean control serum value. To test anti-AIM2 antibody binding in the absence of DNA, TURBO DNase (Life Technologies) was incubated with human sera and AIM2 IVTT products separately at a concentration of 10 U/mL for 20 min at RT prior to the IP reaction being performed. IFI16 antibodies were assayed by ELISA as described (*Matyszewski et al., 2021*). To test antibody binding of the PYRIN domain in the absence of the DNA binding domain, AIM2-PYD was cloned into eGFP expression vector and transiently expressed in 293T cells. Human sera were then used to immunoprecipitate the AIM2 PYD-GFP fusion protein using Protein G Dyna Beads as with the full length AIM2 assay. MNDA was expressed and purified from *E. coli* as previously described (*Matyszewski et al., 2021*) and analyzed by Western blotting using a commercial anti-MNDA antibody (3C1, Cell Signaling). The MNDA ELISA was developed similar to prior ELISA assays (*Antiochos et al., 2021*), and utilized 50 ng of protein per well, blocked with 5% milk in PBS-Tween, and human sera applied at 1:2000 dilution for 2 hr.

## NET assays
Neutrophils were isolated from healthy control PBMCs using Ficoll-Paque density gradient followed by RBC lysis using ACK buffer (Quality Biological). NET formation was induced using PMA at 100 nM for 3 hr. For immunofluorescence studies, neutrophils were plated on glass coverslips for 15 minprior to PMA treatment. For quantitative DNAse protection assays, NETs were induced with PMA in 96 well plates, incubated with or without purified ALRs, then treated with DNAse I at room temperature (RT) prior to incubation with 5 µM Sytox Green (Thermo Fisher) and quantification via fluorimetry using a Perkin Elmer plate reader. Experiments were performed twice.

## Immunofluorescence
Neutrophil samples were stained with anti-MPO-FITC antibody and mounted in DAPI-containing ProLong Gold Antifade Mountant (Thermo Fisher Scientific). AIM2 and IFI16 proteins were expressed,

purified, and fluorescently labeled as previously described (*Morrone et al., 2014*; *Morrone et al., 2015*). SLE renal biopsies were stained as previously described (*Antiochos et al., 2018*) using anti-MPO rabbit polyclonal (ThermoFisher), anti-MPO mouse monoclonal (ThermoFisher), anti-IFI16 mouse monoclonal (Sigma), anti-AIM2 rabbit polyclonal (Sigma), and Hoechst 33,342 (ThermoFisher). Confocal imaging was performed with a Zeiss AxioObserver with 780-Quasar confocal module.

### Fluorescence anisotropy

Fluorescence anisotropy (FA) experiments were performed as in *Matyszewski et al., 2018* to compare DNase shielding of IFI16 versus the catalytic domain of cGAS (cGAS-CD). 150 nM (binding site normalized for each protein) FAM-labeled 72bps dsDNA derived from VACV was pre-incubated for 30 min with 300 nM IFI16 or 500 nM cGAS-CD. DNase I was added at time 0, then the fraction of bound dsDNA was monitored via the FA of dsDNA•protein complex.

### Statistics

Features of patients with and without AIM2 antibodies were compared using Fisher's exact test for categorical variables and the Mann-Whitney test for continuous variables. Multivariable logistic regression was utilized to determine associations between variables. P values less than 0.05 were considered statistically significant.

## Results

### AIM2 autoantibodies are present in SLE, and frequently co-occur with anti-IFI16 and anti-dsDNA antibodies

To determine whether AIM2 was a target of the humoral immune response in SLE, we developed an IP assay to screen for anti-AIM2 antibodies. 41/131 (31.3%) of SLE versus 2/49 (4.1%) of healthy controls were anti-AIM2-positive ($P < 0.001$) (*Figure 1A*). Interestingly, anti-AIM2 antibodies were strongly associated with both anti-IFI16 and anti-DNA antibodies in the SLE samples measured on the day of visit (*Figure 1B* and *Table 1*). We found that anti-AIM2 antibodies were associated with higher measures of SLEDAI ($2.29 \pm 2.3$ vs $1.05 \pm 1.61$, $P = 0.0026$, *Table 1*), which was largely driven by the immunology component. Anti-AIM2 antibodies were associated with the presence of disease activity in the skin domain of the SLEDAI index at the date of blood draw: 11/41 (26.8%) of anti-AIM2 positive patients had scores > 0 in this domain, compared to 11/90 (12.2%) of anti-AIM2 negative patients ($P = 0.0463$). Anti-AIM2 antibodies were also associated with a small but significant increase ($0.63 \pm 0.55$ vs $0.43 \pm 0.51$, $P = 0.0333$) in the SLE Physician Global Disease Activity score, which is based solely on clinical estimation of SLE activity, rather than serologic indices. A multivariable analysis correcting for SLEDAI, anti-dsDNA, and C4 results demonstrated that anti-AIM2 antibodies were significantly associated with anti-IFI16 antibodies with an OR of 3.7 ($P = 0.007$, 95% CI 1.44–9.7). A subset of SLE patients (n = 9) demonstrated particularly high levels of anti-AIM2 antibodies with normalized OD >20 (*Figure 1A*). These patients had higher SLEDAI values than both lower level anti-AIM2-positive and anti-AIM2-negative patients (*Supplementary file 1a*). Among all anti-AIM2 positive patients, we found a higher prevalence of positivity for anti–Ro (18/41, 44% vs 19/90, 21%, $P = 0.0114$) and anti–La (10/41, 24% vs 7/90, 8%, $P = 0.0125$) antibodies (*Supplementary file 1b*).

SS shares several phenotypic features with SLE, including the presence of an IFN signature and B cell dysregulation (*Yao et al., 2013*), but anti-DNA antibodies are not characteristic of SS. We therefore analyzed SS sera for the presence of anti-AIM2 antibodiesand found that 46/133 (34.6%) of SS sera were positive. In contrast to SLE, anti-IFI16 was not enriched in patients with anti-AIM2 antibodies in SS (35% anti-AIM2-positive and anti-IFI16-positive versus 28% anti-AIM2-negative and anti-IFI16-positive in SS, $P = 0.4324$), showing that the association between anti-IFI16 and anti-AIM2 antibodies is specific to SLE, where these immune responses are also associated with anti-dsDNA antibodies.

Because AIM2 binds dsDNA of any sort, we considered whether the IP reaction observed here might identify interactions between AIM2 and dsDNA bound in circulating SLE immune complexes (*Means et al., 2005*), rather than a direct antibody-AIM2 interaction. To address this, we pre-treated anti-AIM2 +SLE sera and the AIM2 IVTT product with DNase prior to combining them in the IP reaction, and found that SLE sera retained the ability to bind AIM2 following DNase treatment (*Figure 1C–D*). To further confirm this, we also performed immunoprecipitation of AIM2 PYRIN domain (PYD) that lacks

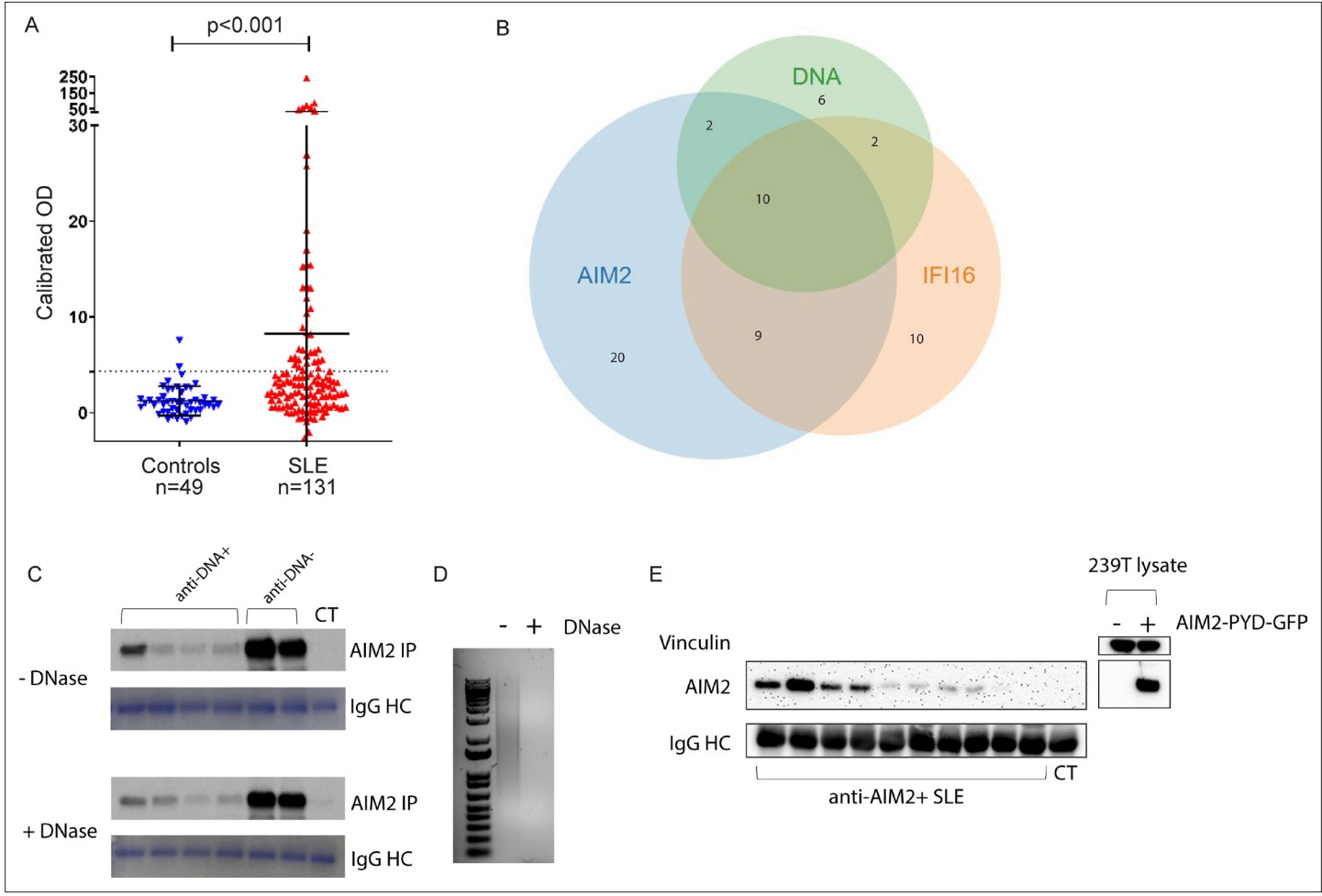

**Figure 1.** Anti-AIM2 antibodies are associated with anti-IFI16 and anti-DNA antibodies in SLE. AIM2 antibodies were detected using immunoprecipitation (IP) of $^{35}$S-methionine labeled, in vitro transcribed and translated protein. Data are presented as OD units calibrated to a known positive reference serum. Dotted line indicates positive threshold value determined as the mean + 2 standard deviations of control serum samples. AIM2 autoantibodies were identified in 2/49 controls and 41/131 SLE patients. Statistical significance was determined using the Mann-Whitney test for nonparametric values (**A**). Relationship between anti-AIM2, -IFI16, and –DNA antibodies in the SLE cohort (**B**). Anti-AIM2 +SLE and control (CT) sera and AIM2 protein were each treated with or without DNase prior to being combined in the IP reaction. Coomassie stain of IgG heavy chain (HC) is shown below each IP result (**C**). 1 µg of Poly(dA:dT) was treated with DNase as in (**C**) and analyzed by SYBR Green staining in agarose gel (**D**). 293T cells were transfected with AIM2-PYD-GFP expression plasmid, and lysate was used in IP reaction with anti-AIM2 +SLE and CT sera (**E**). IP products and 293T lysates were blotted for AIM2 using anti-N terminal antibody (Cell Signaling D5X7K).

The online version of this article includes the following figure supplement(s) for figure 1:

**Figure supplement 1.** MNDA autoantibodies are not enriched in SLE.

the DNA-binding HIN domain, finding that 8/10 anti-AIM2 +SLE sera immunoprecipitated the isolated AIM2 PYD in this assay. Thus, DNA was not required for AIM2 binding in the samples assessed in these assays, confirming that this is a bona fide antibody specificity. It remains possible that the presence of dsDNA in SLE immune complexes could theoretically enhance AIM2 binding in vivo, with important consequences for epitope spreading and amplification of inflammation in tissues (see below).

The identification of anti-AIM2 autoantibodies raises the question of whether additional ALR proteins might be SLE autoantigens. To address this point, we developed an ELISA against MNDA – a third member of the ALR dsDNA sensor family. Despite the fact that antibodies against peptides derived from MNDA had been reported in some patients with SLE and other rheumatic diseases (*van Beers et al., 2013*), we did not identify any difference in anti-MNDA reactivity between healthy control and SLE sera (median AU HC = 1.279, median AU SLE = 1.228, *P* = 0.9319) (*Figure 1—figure supplement 1*).

**Table 1.** Day of visit phenotypic characteristics of SLE patients related to AIM2 autoantibody status.

| Autoantibody | Anti-AIM2+ n = 41 | Anti-AIM2- n = 90 | P value |
|---|---|---|---|
| IFI16 Positive | 19/41 (46%) | 12/90 (13%) | < 0.0001 |
| DNA Positive | 12/41 (29%) | 7/89 (8%) | 0.0026 |
| **Disease Activity Feature** | | | |
| Physician Global Disease Activity | 0.63 ± 0.55 | 0.43 ± 0.51 | 0.0333 |
| SLEDAI | 2.29 ± 2.3 | 1.05 ± 1.61 | 0.0026 |
| C3 | 114.7 ± 36.9 | 121.4 ± 29.0 | 0.1352 |
| C4 | 19.5 ± 8.2 | 25 ± 9.3 | 0.0005 |
| Urine Protein/Creatinine ratio | 0.134 ± 0.15 | 0.107 ± 0.11 | 0.4372 |

Numerators correspond to number of patients with indicated feature positive and denominators to total number of patients with indicated feature recorded in the cohort, followed by percent (%) positive.

## AIM2 and IFI16 bind to neutrophil extracellular traps and inhibit their degradation by DNase I

The close relationship between anti-AIM2, anti-IFI16, and anti-dsDNA antibodies in the SLE cohort led us to consider scenarios in which ALR-DNA complexes could be generated and promote the development of autoantibodies against these three antigens. Neutrophil extracellular traps (NETs) have been implicated as important sources of extracellular DNA in SLE and are linked to the IFN signature as well as autoantibody generation in this disease (*Gupta and Kaplan, 2016*). ALRs are IFN-induced, bind to dsDNA of many origins in a sequence-independent manner, and AIM2 has been identified as a protein constituent of SLE NETs in a proteomics analysis (*Bruschi et al., 2019*). IFI16 is released from epithelial cells undergoing apoptosis (*Antiochos et al., 2018*; *Costa et al., 2011*), and extracellular IFI16 is quantifiable in the sera of SLE patients (*Gugliesi et al., 2013*). Considering these observations, we reasoned that when ALRs are generated in the setting of IFN exposure and subsequently released from cells, they might encounter and bind to extracellular NETs, accumulating on this extracellular platform and creating a hub for amplification similar to that observed in the complement and coagulation pathways (*de Bont et al., 2019*).

To test this hypothesis, we used NETs as a DNA substrate for ALR binding: neutrophils were stimulated to undergo NETosis with PMA, and then incubated with fluorescently labeled IFI16 and AIM2 proteins. We found that both ALRs bind readily to NETs (*Figure 2A–B*). Co-localization of AIM2 and IFI16 along NET chromatin fibers was visible by confocal microscopy in this analysis, suggesting that both ALRs assemble into filaments on NET DNA.

IFI16 and AIM2 nucleoprotein filaments are highly stable and persist even after the dsDNA template has been degraded by nucleases (*Antiochos et al., 2018*; *Matyszewski et al., 2018*), and impaired clearance of NETs by DNase I has been observed in some SLE patients (*Hakkim et al., 2010*). We therefore hypothesized that ALRs might represent one group of proteins with the ability to mediate NET resistance to nucleases, potentially enhancing antigenicity. We used DNase I to explore this question, as DNase I is the nuclease responsible for effective clearance of NETs (*Hakkim et al., 2010*), and DNase I deficiency has been associated with SLE in both human subjects and animal models (*Martínez Valle et al., 2008*). After exposure to 20 U/ml DNase I for 1 hr at RT, both myeloperoxidase (MPO) and DNA signals were completely degraded, leaving no observable fluorescence in any channel (*Figure 2C*). When NETs were first incubated with ALRs, however, we observed incomplete ALR-NET degradation by DNase I – in some areas, IFI16 and AIM2 remained present and co-localized with MPO (*Figure 2D*). In addition, there was observable DNA remaining in these areas of persistent ALR structures, implying that the ALRs had partially shielded NET DNA from degradation. This finding suggested that both the protein and DNA components of the ALR-NET structure are resistant to DNase-mediated clearance. To better quantify this, we employed a plate-based Sytox Green assay to measure the dsDNA content of NETs following exposure to DNase I (*Figure 2E*). This assay confirmed that ALR-bound NETs are resistant to DNase I, leaving more DNA present following

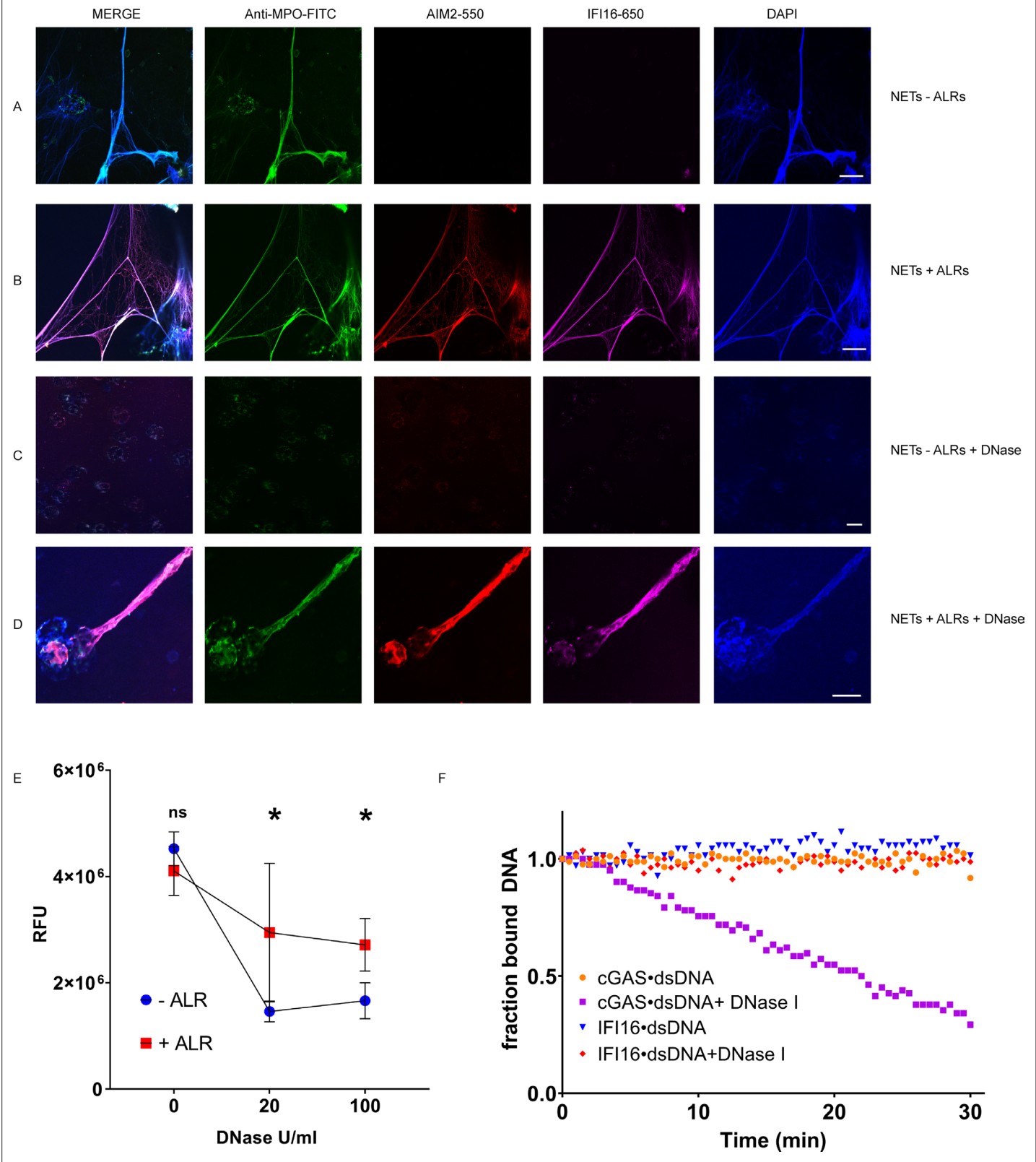

**Figure 2.** IFI16 and AIM2 bind NETs and prevent NET degradation by DNase I. NETs were induced in neutrophils using PMA 100 nM for 3 hr, then left untreated (**A**) or incubated with fluorescently labeled AIM2 (pink) and IFI16 (red) at 200 nM at RT for 1 hr (**B**). Following ALR incubation, samples were stained with anti-MPO-FITC antibody (green) and DAPI (blue), then imaged by confocal microscopy. NETs were treated with DNase I at 20 U/mL at RT for 1 hr (**C**). NETs incubated with ALRs as in (**B**) were then treated with 20 U/mL DNase I for 1 hr (**D**). Scale bars = 20 μm. NETs in 96 well plates

*Figure 2 continued on next page*

*Figure 2 continued*

were incubated with ALRs at 200 nM (or buffer only) for 1 hr at RT, then treated with DNase I at 0, 20, and 100 U/mL for 30 min at RT. NETs were then stained with Sytox-Green 5 µM, and samples analyzed by fluorimetry (**E**). RFU = fluorescence units. Mean and standard deviation of 4 replicate wells are indicated. Mann-Whitney test was used to compare groups. $P > 0.05$ = not significant (ns). $P < 0.05$ = significant (*). IFI16 and the catalytic domain of cGAS were combined with FAM-labeled 72 bp VACV dsDNA for 30 min, then DNase I added at concentration of 20 U/mL at time = 0 and the fraction of bound dsDNA was monitored via the fluorescence anisotropy of dsDNA•protein complex (**F**).

nuclease treatment (*Figure 2E*). Together these experiments demonstrate that ALRs bind to NETs, generating a protein-DNA structure with enhanced resistance to DNase-mediated clearance.

To test the specificity of this observation, we utilized a fluorescence anisotropy assay to compare IFI16 against the catalytic domain of cGAS (cGAS-CD), to determine whether the property of DNA binding alone confers proteins with the ability to shield bound dsDNA from DNAse I (*Figure 2F*). In this second assay, we again observed that IFI16 blocked dsDNA degradation, while cGAS-CD did not. Thus, the ability to shield dsDNA from nuclease does not appear to be a universal property of all DNA-binding proteins, and other factors (such as the ability to oligomerize) may be required for this behavior.

## IFI16-NETs are present in lupus nephritis

Prior studies have presented evidence of in vivo NET formation within the renal tissues of SLE patients, supporting the notion that dysregulated neutrophil function contributes to immune pathology in this disease (*Villanueva et al., 2011*). We therefore sought to determine whether ALR-NET interactions could be identified among NETs present in lupus nephritis biopsies. Considering that patients with diffuse proliferative lupus nephritis are known to harbor netting neutrophils in renal tissue (*Villanueva et al., 2011*), we identified patients with diffuse proliferative lupus nephritis, then selected 5 samples whose biopsies demonstrated neutrophilic infiltrates or karyorrhectic debris (*Supplementary file 1c*). We found that AIM2 was highly expressed in MPO-positive infiltrating cells (*Figure 3A*), while IFI16 was expressed more broadly throughout renal cell types (*Figure 3B*). We detected NETs containing both AIM2 and IFI16 in glomerular and interstitial infiltrates (*Figure 3C and D*). High magnification, z-stack imaging (*Figure 3—figure supplement 1*) confirmed that these structures represented extra-cellular DNA that co-stained for MPO and AIM2 or IFI16, consistent with ALR-bound NETs, rather than adjacent or overlapping cell nuclei. In summary, our immunostaining experiments provide evidence that AIM2 and IFI16 bind NETs in the setting of diffuse proliferative lupus nephritis, establishing AIM2 and IFI16 as NET-bound SLE autoantigens.

## Discussion

SLE features autoantibodies that bind nucleic acids and nucleic acid-binding proteins, and extra-cellular nucleic acids contribute to SLE pathogenesis (*Mustelin et al., 2019*). Here, we identify the dsDNA sensor AIM2 as a novel autoantigen in SLE and demonstrate that anti-AIM2 antibodies are associated with SLE disease activity markers. Furthermore, we find that NETs provide a scaffold for ALR oligomerization, which in turn confers resistance to nuclease degradation.

NETosis is a process whereby dsDNA is expelled into the extracellular space at sites of tissue damage and is of mechanistic relevance in SLE (*Gupta and Kaplan, 2016*). The NET dsDNA scaffold is a structure on which a variety of molecules can interact, and is a source of antigenic proteins in SLE and other inflammatory diseases (*Mistry and Kaplan, 2017*; *Lee et al., 2017*). While the ALRs have been most extensively studied for their intracellular functions, there are reasons to theorize that they may perform extracellular activities as well. Our previous work demonstrated that IFI16 is released from epithelial cells injured by the NK cell granule killing pathway and that IFI16 retains its ability to interact with dsDNA in the extracellular space (*Antiochos et al., 2018*). Our findings are consistent with the observations that (i) IFI16 is released from irradiated epithelial cells, and (ii) extracellular IFI16 is quantifiable in the sera of SLE patients (*Gugliesi et al., 2013*; *Iannucci et al., 2020*). In addition, the release of AIM2 from pyroptotic cells into extracellular vesicles in the context of pyroptosis has been described (*So et al., 2018*) and extracellular signaling activities of the related oligomerizing inflammasome adaptor protein ASC have also been reported (*Franklin et al., 2014*). We therefore reasoned that ALRs released from injured epithelial, endothelial, or other immune cells, generated in the setting

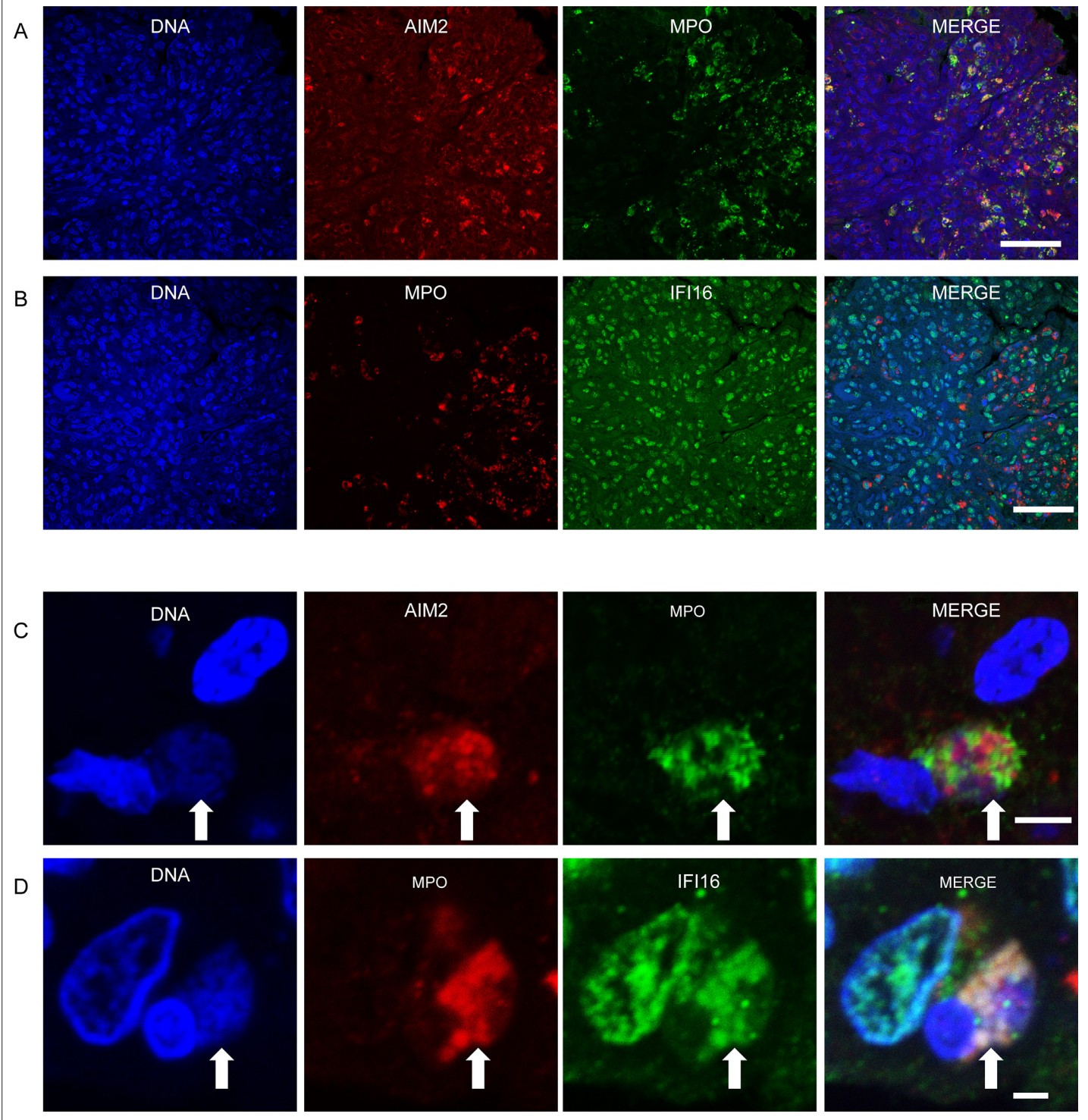

**Figure 3.** IFI16 and AIM2 bind NETs in diffuse proliferative lupus nephritis. Representative images of ALR expression and ALR-NETs identified in patients with class IV lupus nephritis. AIM2 (**A**) expression was largely detected in MPO expressing neutrophils, while IFI16 (**B**) was more broadly distributed. NETs (arrows) demonstrating co-localizing staining for DNA, MPO, and AIM2 (**C**) or IFI16 (**D**) visualized by confocal microscopy. Scale bars: 50 µm (**A, B**) 5 µm (**C**), 2 µm (**D**).

The online version of this article includes the following figure supplement(s) for figure 3:

**Figure supplement 1.** Z-stack imaging of AIM2/IFI16-NETs in lupus nephritis.

of SLE end-organ tissue damage, might encounter and bind to extracellular NETs. We found that both IFI16 and AIM2 readily assemble into filaments along the length of NET dsDNA. Unexpectedly, we found this ALR-NET structure resists DNase-mediated degradation – a property that was not demonstrated by cGAS-CD. NETs promote IFN signaling at sites of their generation when engulfed by immune cells (*Garcia-Romo et al., 2011*; *Apel et al., 2021*), and may have additional disease-amplifying functions (*Salazar-Gonzalez et al., 2019*). By prolonging the stability of interferogenic NETs, extracellular ALRs may enhance IFN signaling at sites of neutrophil activation, which could be further amplified by IFN-induced expression of the ALRs themselves.

Impairment of NET removal has been specifically linked to the presence of lupus nephritis (*Hakkim et al., 2010*) - a manifestation of SLE with significant associated morbidity (*Gasparotto et al., 2020*). Neutrophilic infiltration of the kidney is a feature of more severe forms of glomerulonephritis, and NET formation in this organ may contribute to renal damage through the propagation of IFN signaling, immune cell activation, and thrombosis (*Salazar-Gonzalez et al., 2019*). Confocal microscopy has been utilized to demonstrate the presence of NET structures in renal lesions of patients with SLE (*Villanueva et al., 2011*; *Frangou et al., 2019*) and also ANCA associated vasculitis (*Kessenbrock et al., 2009*), supporting the notion that NETs play a pathogenic role in the immune dysregulation and tissue damage that occur in glomerulonephritis.

Here, we demonstrate for the first time that the DNA sensors AIM2 and IFI16 bind to NETs in vivo, through imaging studies of proliferative lupus nephritis specimens. Our data include z-stack images at high magnification, clearly demonstrating the presence of extracellular DNA-MPO-ALR complexes in this site. This finding supports previous data (*Gugliesi et al., 2013*; *Iannucci et al., 2020*; *Bawadekar et al., 2015*) suggesting that the ALRs may have important functions not just intracellularly, but also in the extracellular environment. The large chromatin fibers generated through NETosis represent sizeable dsDNA templates upon which IFI16 and AIM2 monomers oligomerize in the extracellular space and are expected to result in durable immunostimulatory structures at sites of IFN-induced protein expression. ALR-bound NETs may therefore promote not only local immune activation but the targeting of ALRs (and DNA) by antibodies in SLE. NET-bound ALRs may potentiate autoimmunity by activating B cells through either endosomal TLR9-NET DNA interactions, or activation of the B cell receptor, in a manner analogous to that reported for the NET-binding peptide LL37 (*Gestermann et al., 2018*). Alternatively, the initial presence of anti-DNA antibodies containing Ig-bound DNA (derived from NETs or other sources) could lead to immune complexes consisting of Ig-DNA-ALR interactions in the periphery, thereby promoting the subsequent development of anti-ALR antibodies through epitope spreading.

Our data indicate that AIM2 is targeted not only in SLE but also in SS – a condition in which NETosis has not been linked to disease pathology. In contrast to the relationship seen in SLE, we found no association between AIM2 and IFI16 antibodies in SS, and anti-dsDNA antibodies are absent in SS. This difference highlights the important role of disease-specific tissue processes in the development of unique autoantibody profiles against shared antigens. In the setting of lupus nephritis, we observed ALR expression by both neutrophils and resident renal cells and suspect that these antigens may be released by a variety of cell types in the kidney, leading to the observed extracellular interaction with NET DNA. Contrastingly, neutrophil infiltration in target salivary tissues is not a common feature of SS, and the absence of the NET DNA scaffold may explain the differing autoantibody profile observed in that condition.

In summary, we have identified the ALRs AIM2 and IFI16 as NET-binding autoantigens in SLE. The ALR-NET interaction may increase NET longevity and perpetuate NET-mediated inflammatory signaling in lupus nephritis and other sites of NET generation and IFN expression. This work supports a role for the ALRs in extracellular immune processes as NET-binding antigens in SLE.

## Additional information

### Funding

| Funder | Grant reference number | Author |
|---|---|---|
| National Institute of Arthritis and Musculoskeletal and Skin Diseases | K08AR077100 | Brendan Antiochos |
| National Institute of Arthritis and Musculoskeletal and Skin Diseases | K23AR075898 | Christopher Mecoli |
| Jerome L. Greene Foundation | | Brendan Antiochos Christopher Mecoli |
| National Institutes of Health | R01 GM 129342 | Jungsan Sohn |
| National Institute of Arthritis and Musculoskeletal and Skin Diseases | AR069572 | Michelle Petri |
| National Institute of Arthritis and Musculoskeletal and Skin Diseases | R01 DE12354 | Antony Rosen |

The funders had no role in study design, data collection and interpretation, or the decision to submit the work for publication.

### Author contributions

Brendan Antiochos, Conceptualization, Formal analysis, Funding acquisition, Investigation, Writing - original draft, Writing - review and editing; Daniela Trejo-Zambrano, Archit Garg, Investigation, Methodology; Paride Fenaroli, Avi Rosenberg, Alan Baer, Michelle Petri, Daniel W Goldman, Resources, Writing - review and editing; Jungsan Sohn, Resources, Supervision, Writing - review and editing; Jessica Li, Christopher Mecoli, Formal analysis, Methodology, Writing - review and editing; Livia Casciola-Rosen, Antony Rosen, Conceptualization, Resources, Supervision, Writing - review and editing

### Author ORCIDs

Brendan Antiochos http://orcid.org/0000-0001-6166-9750
Jungsan Sohn http://orcid.org/0000-0002-9570-2544
Livia Casciola-Rosen http://orcid.org/0000-0002-4172-1539

### Ethics

Human subjects: Informed consent was obtained from all human subjects enrolled in the SLE, Sjogren's and healthy control cohorts. Approval for the involved cohort studies was obtained from the Johns Hopkins IRB (study numbers NA_00039294 and NA_00013201).

### Decision letter and Author response

Decision letter https://doi.org/10.7554/eLife.72103.sa1
Author response https://doi.org/10.7554/eLife.72103.sa2

## Additional files

### Supplementary files

• Supplementary file 1. AIM2 autoantibody correlations and renal biopsy findings in SLE cohort. (a) Phenotypic Characteristics of SLE Patients Related to AIM2 Autoantibody Level. Numerators correspond to number of patients with indicated feature positive and denominators to total number of patients with indicated feature recorded in the cohort, followed by percent (%) positive.

(b) Immunologic phenotype of SLE patients related to AIM2 autoantibody status. Numerators correspond to number of patients with indicated feature positive and denominators to total number of patients with indicated feature, followed by percent (%) positive.

• Transparent reporting form

## Data availability

All data generated or analyzed during this study are included in the manuscript and supporting files.

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
