## [Editor Report]

This paper identifies proteins that serve as targets of self-reactive autoantibodies during an autoimmune disease called systemic lupus erythematosus (commonly referred to as lupus). Importantly, the authors provide evidence that these proteins bind and protect extracellular DNA from destruction, and propose that this property may enhance the autoimmune response to the DNA and associated proteins. The work may therefore provide an important underlying mechanism for a prevalent and important human autoimmune disease.

---

## [Decision Letter]

**Decision letter after peer review:**

Thank you for submitting your article "The DNA sensors AIM2 and IFI16 are NET-binding SLE autoantigens" for consideration by *eLife*. Your article has been reviewed by 3 peer reviewers, and the evaluation has been overseen by a Reviewing Editor and Betty Diamond as the Senior Editor. The following individual involved in review of your submission has agreed to reveal their identity: Kate Fitzgerald (Reviewer #1).

Essential revisions:

1) Reviewer 2 raises an important issue as to the specificity of the assay measuring anti-Aim2 autoantibodies. The concern is specifically that the assay detects other antibodies (e.g. anti-DNA antibodies) that bind to a complex (e.g. containing DNA) to which the AIM2 protein binds as well. The reviewers discussed this concern and agreed that it should be addressed. One suggestion was to repeat the experiment with a radiolabelled AIM2 protein that lacks a DNA binding domain. Alternatively, the presence of anti-AIM2 antibodies could be assessed with another technique, e.g. immunoblotting. Regardless, some additional experimentation appears to be required to solidify this central claim of the manuscript.

2) Reviewers 1 and 3 raise other issues that can be discussed by the authors. It would be ideal if there were data to address the specificity of the effects (e.g. comparison to other DNA binding proteins), but the authors are welcome to try to address the point by discussion if they feel that the concern would be satisfactorily addressed in this manner.

*Reviewer #1 (Recommendations for the authors):*

Overall the data is compelling and convincing. A few issues should be addressed.

1. It would be useful to address the specificity of these responses. For example are autoantibodies to other ALRs or DNA sensors (eg. cGAS) also feature of SLE or is there something unique about these 2 ALR family members.

2. The ability of AIM2 or IFI16 to bind NETs and prevent their destruction by Dnase-1 could be controlled better. Do other dsDNA binding proteins similarly shield NETs from destruction or is this unique to ALRs.

3. The authors should discuss in more detail how AIM2 and IFI16 accumulate extracellularly in SLE patient samples to modulate the stability and interferogenic potential of NETs.

*Reviewer #2 (Recommendations for the authors):*

The authors of Antiochos et al., present important observations of the behavior of AIM2 in association with NET DNA and in lupus kidney biopsies. It is observed that AIM2 has a strong affinity for NET DNA and that it appears to be expressed by neutrophils. This may provide an opportunity for AIM2 to escape the cell confines and become exposed to the external milieu, in which it may contribute to a immune stimulation by nuclear antigens. If so, the high affinity for DNA and AIM2's ability to form larger assemblies on the DNA may explain why the binding of AIM2 to DNA NETs makes them resistant to DNAse 1 digestion.

However, this observation also provides the major concern for the detection of anti-AIM2 autoantibodies. Lupus plasma may contain anti-DNA antibodies and extracellular DNA (either free or in complex with antibodies). Such DNA-Ig complexes could then act as a bridge to capture the recombinant AIM2 during the IP procedure, even without the direct participation of anti-AIM2 antibodies. Unfortunately, there may not be a simple way to unambiguously discriminate between these alternatives. Treating the plasma with DNAse 1 may not disrupt the DNA-AIM2 complexes, as shown in Figure 2. Perhaps a Western blot to the purified AIM2 could be attempted, in at least a subset of plasmas that are either anti-DNA positive or negative?

*Reviewer #3 (Recommendations for the authors):*

Potential explanations for the immunogenicity of NET-bound proteins should be addressed. For example, does the increased valency of multimerized antigen result in greater activation through the BCR, or does does association with DNA result in the activation of additional pattern recognition receptors, or can sensor/DNA complexes continue to signal either extracellularly or when taken up by scavenger cells.

---

## [Author Response]

Essential revisions:1) Reviewer 2 raises an important issue as to the specificity of the assay measuring anti-Aim2 autoantibodies. The concern is specifically that the assay detects other antibodies (e.g. anti-DNA antibodies) that bind to a complex (e.g. containing DNA) to which the AIM2 protein binds as well. The reviewers discussed this concern and agreed that it should be addressed. One suggestion was to repeat the experiment with a radiolabelled AIM2 protein that lacks a DNA binding domain. Alternatively, the presence of anti-AIM2 antibodies could be assessed with another technique, e.g., immunoblotting. Regardless, some additional experimentation appears to be required to solidify this central claim of the manuscript.2) Reviewers 1 and 3 raise other issues that can be discussed by the authors. It would be ideal if there were data to address the specificity of the effects (e.g., comparison to other DNA binding proteins), but the authors are welcome to try to address the point by discussion if they feel that the concern would be satisfactorily addressed in this manner.

Dr. Fitzgerald (Reviewer 1) raises the issue of the specificity of the antibody response to IFI16 /AIM2 as opposed to other ALRs or dsDNA sensors. To address this important point, we developed an ELISA against MNDA – another ALR protein with DNA-binding function. We utilized this assay to compare SLE and control sera, and did not identify a difference in antibody response against MNDA in these two groups (Added as Supplemental Figure 1), indicating that MNDA is not a SLE-associated autoantigen. We conclude that autoantigen status is not a universal feature of the ALRs, but still think it plausible that additional dsDNA sensors may be identified as novel autoantigens. (Please see: Methods (Lines 108112); Results (Lines 177-182); Figure 1—figure supplement 1).

Dr. Fitzgerald also points out that the ability to shield dsDNA from DNase may not be a unique property of these ALRs, in that other DNA-binding proteins could share this ability to interfere with nucleasemediated degradation. To address this, we utilized fluorescence anisotropy to compare IFI16 with the cGAS catalytic domain (added as Figure 2 F) with respect to DNase shielding ability. While IFI16 again showed the ability to resist DNase in this assay, DNA bound to cGAS-CD was readily degraded following the addition of DNase I to the reaction. This additional data demonstrates that the ability to interfere with DNase activity is not necessarily a property of any protein – rather, that the formation of higher order oligomers along dsDNA (a property of ALRs) might be unique in this capacity to resist DNase I.

(Please see: Methods (Lines 127-131); Results (Lines 220-225); Figure 2 F).

In her review, Dr. Fitzgerald also suggests additional discussion of the concept of ALR activity in the extracellular environment. We have added several references to the discussion that further supports this concept: (i) our own work that showed IFI16 release from epithelial cells exposed to cytotoxic granules, (ii) work by Gugliesi et al., and Iannucci et al., that demonstrated IFI16 release from irradiated cell lines, and also quantified extracellular IFI16 in sera of SLE patients (and other rheumatic diseases), (iii) the observation by So et al., that AIM2 is transported extracellularly (in exosomes) from cells undergoing pyroptosis, (iv) work by Franklin et al., demonstrating that the oligomerizing inflammasome adaptor ASC is also released into the extracellular space by activated cells, where is retains signaling function. This addition to the discussion appears in lines 251-261.

Reviewer #2 (Recommendations for the authors):The authors of Antiochos et al., present important observations of the behavior of AIM2 in association with NET DNA and in lupus kidney biopsies. It is observed that AIM2 has a strong affinity for NET DNA and that it appears to be expressed by neutrophils. This may provide an opportunity for AIM2 to escape the cell confines and become exposed to the external milieu, in which it may contribute to a immune stimulation by nuclear antigens. If so, the high affinity for DNA and AIM2's ability to form larger assemblies on the DNA may explain why the binding of AIM2 to DNA NETs makes them resistant to DNAse 1 digestion.However, this observation also provides the major concern for the detection of anti-AIM2 autoantibodies. Lupus plasma may contain anti-DNA antibodies and extracellular DNA (either free or in complex with antibodies). Such DNA-Ig complexes could then act as a bridge to capture the recombinant AIM2 during the IP procedure, even without the direct participation of anti-AIM2 antibodies. Unfortunately, there may not be a simple way to unambiguously discriminate between these alternatives. Treating the plasma with DNAse 1 may not disrupt the DNA-AIM2 complexes, as shown in Figure 2. Perhaps a Western blot to the purified AIM2 could be attempted, in at least a subset of plasmas that are either anti-DNA positive or negative?

An essential revision suggested by Dr. Fitzgerald and Reviewer 2 surrounds the notion that DNA-binding proteins might interact with SLE immune complexes indirectly, through DNA already present in circulating immune complexes. This is indeed an important point and to address it we used a subset of SLE sera (both anti-DNA positive and negative) in an immunoprecipitation (IP) reaction that was preceded by DNase treatment. We incubated both the patient serum and the AIM2 protein with DNase separately to minimize any accessible DNA from either source, prior to combining them in the IP reaction (added as Figure 1 C-D). We used a DNase pre-treatment that was adequate to remove a large quantity of dsDNA – at least one microgram, as shown in Figure 1 D. This did not affect the ability of AIM2 autoantibodies to IP AIM2. Reviewer 2 also points out that DNA-bound proteins (like the very ALRs we report on) could potentially shield Ig-bound DNA from DNase in this reaction. While this could potentially occur, DNA that is effectively shielded from DNase due to binding by another protein seems unlikely to be accessible to AIM2, given the length-dependency of AIM2-DNA binding. AIM2 shows minimal affinity for 10 bp DNA templates (representing a binding site of one AIM2 monomer), and requires longer stretches of dsDNA to form stable oligomers (1). It therefore seems unlikely that DNA held in an Ig-protein-DNA complex could be simultaneously protected from DNase I yet accessible to AIM2 in this IVTT assay. In addition, the DNase was not removed prior to the IP reaction, but rather remained in solution after sera and AIM2 were combined. If any Ig-bound DNA that survived DNase treatment were later released, it would likely be accessible to nuclease digestion during the assay. Nonetheless, to further probe this issue, and as suggested by the Editor, we examined the ability of anti-AIM2+ SLE sera to immunoprecipitate the AIM2 pyrin domain (PYD) alone, in a construct lacking the DNA-binding HIN domain. We expressed the AIM2-PYD using an eGFP expression plasmid in 293T cells, and performed IP using 10 SLE anti-AIM2+ sera, finding that 8 of the 10 sera immunoprecipitated the PYD alone in this assay (Figure 1E). We note that it is possible that human sera may target epitopes in select sites of the molecule, and a negative result for a minority of sera in this assay may simply indicate that a given serum binds epitopes in the HIN domain. Together, these additional data address the critique regarding a possible indirect Ig-DNA-AIM2 interaction being primarily detected by this IVTT IP assay. Importantly though, we do consider it plausible that such an interaction between DNA in immune complexes and ALRs (or other high affinity dsDNA binding proteins) might occur in vivo, and we wonder whether this mechanism might in fact contribute to the generation of anti-AIM2 antibodies in some cases through intermolecular epitope spreading facilitated by extracellular dsDNA. (Please see Methods (Lines 102-108); Results (Lines 166-176); Figure 1 C,D, E). Taken together, we believe that these revisions answer a key criticism of the prior review.

Reviewer #3 (Recommendations for the authors):Potential explanations for the immunogenicity of NET-bound proteins should be addressed. For example, does the increased valency of multimerized antigen result in greater activation through the BCR, or does does association with DNA result in the activation of additional pattern recognition receptors, or can sensor/DNA complexes continue to signal either extracellularly or when taken up by scavenger cells.

Reviewer 3 notes the relevant reference by Hakkim et al., who reported the phenomenon of impaired NET degradation by DNase I in some patients with SLE – we have cited this reference in the Results section (line 204) and in the discussion (268).

Reviewer 3 also points out the salient point that NET-bound proteins may demonstrate immunogenicity through multiple mechanisms – we share this author’s view that numerous such properties or mechanisms may contribute to this phenomenon. In the discussion, we have added reference to the observation that the NET-binding peptide LL37 has shown the ability to promote TLR9-mediated signaling and also B cell receptor activation, and we theorize that other DNA-binding autoantigens such as the ALRs could demonstrate similar behavior. This discussion is now included in lines 283-289.